# In Situ DRIFTS-MS Methanol Adsorption Study onto Supported NiSn Nanoparticles: Mechanistic Implications in Methanol Steam Reforming

**DOI:** 10.3390/nano11123234

**Published:** 2021-11-28

**Authors:** Luis F. Bobadilla, Lola Azancot, Svetlana Ivanova, Juan J. Delgado, Francisca Romero-Sarria, Miguel A. Centeno, Anne-Cécile Roger, José A. Odriozola

**Affiliations:** 1Departamento de Química Inorgánica e Instituto de Ciencia de Materiales de Sevilla, Centro Mixto CSIC-Universidad de Sevilla, 49 Américo Vespucio, 41092 Sevilla, Spain; lola.azancot@icmse.csic.es (L.A.); sivanova@us.es (S.I.); francisca@us.es (F.R.-S.); centeno@icmse.csic.es (M.A.C.); odrio@us.es (J.A.O.); 2Departamento de Ciencia de los Materiales e Ingeniería Metalúrgica y Química Inorgánica, Facultad de Ciencias, Universidad de Cádiz, Campus Río San Pedro, 11510 Puerto Real (Cádiz), Spain; juanjose.delgado@uca.es; 3Institut de Chimie et Procédés pour l’Energie, l’Environnement et la Santé, ICPEES-ECPM, UMR CNRS 7515, Université de Strasbourg, 25 Rue Becquerel, 67807 Strasbourg, France; annececile.roger@unistra.fr; 4Department of Chemical and Process Engineering, University of Surrey, Guildford GU2 7XH, UK

**Keywords:** NiSn nanoparticles, methanol steam reforming, mechanism, in situ DRIFTS-MS

## Abstract

Methanol adsorption over both supported NiSn Nps and analogous NiSn catalyst prepared by impregnation was studied by in situ diffuse reflectance infrared Fourier transform spectroscopy (DRIFTS) to gain insights into the basis of hydrogen production from methanol steam reforming. Different intermediate species such as methoxides with different geometry (bridge and monodentate) and formate species were identified after methanol adsorption and thermal desorption. It is proposed that these species are the most involved in the methanol steam reforming reaction and the major presence of metal-support interface sites in supported NiSn Nps leads to higher production of hydrogen. On the basis of these results, a plausible reaction mechanism was elucidated through the correlation between the thermal stability of these species and the evolution of the effluent gas released. In addition, it was demonstrated that DME is a secondary product generated by condensation of methoxides over the acid sites of alumina support in an acid-catalyzed reaction.

## 1. Introduction

With the depletion of fossil fuels and the growing global energy demand, as well as the Paris agreement achieved in the 21st Conference of Parties in 2015, to reduce the emission of greenhouse gases, it becomes essential to substitute the traditional energy and transport sectors based on crude oil, natural gas and coal by green fuels and low-carbon energy systems [1].

Hydrogen represents a versatile energy vector for deep decarbonization that has attracted great interest due to potential applications on proton-exchange membrane fuel cells (PEMFC’s). Hydrogen can be produced from renewable sources such as biomass, wastes and water. Although novel technologies based on photocatalytic processes are emerging [2,3], hydrogen is still mainly generated by thermal processes. Due to its high energy density, low price and easy transportation, a growing interest in methanol as a fuel for hydrogen production has been concerned in the past two decades [4]. Hydrogen is essentially produced by methanol steam reforming according to Equation (1). This overall reaction is accompanied by other side reactions such as methanol decomposition (Equation (2)) and water gas-shift reaction (Equation (3)), as follows [5]:*Methanol steam reforming*: CH_3_OH + H_2_O ↔ CO_2_ + 3H_2_ ∆H_0_ = +49.7 kJ mol^−1^(1)
*Methanol decomposition*: CH_3_OH ↔ CO + 2H_2_ ∆H_0_ = +92.0 kJ mol^−1^(2)
*Water gas-shift reaction*: CO + H_2_O ↔ CO_2_ + H_2_ ∆H_0_ = −41.2 kJ mol^−1^(3)

Traditionally, Cu-based catalysts have been the most common ones used for methanol steam reforming due to their high activity and low cost [6]. The performance of Cu-based catalysts depends on the copper exposed area and the metallic dispersion, and thus the preparation method directly affects the catalytic activity and the properties of the Cu catalysts [7,8,9]. Furthermore, numerous studies have demonstrated that the addition of promoters such as Zn or Mn decrease the particle size of copper and enhance notably the catalytic performance of Cu-based catalysts [10,11,12]. For instance, Dasireddy et al. [13] reported recently an interesting study based on Cu–Mn–O systems that present a high time-on-stream conversion of methanol under reforming conditions. Due to their low-cost, these materials are very attractive for renewable hydrogen production technology to the commercial levels. Moreover, indium is demonstrated to be an effective promoter in Cu-based catalysts for H_2_ production in steam reforming of methanol [14]. However, Cu-based catalysts are known for their pyrophoric properties and the deactivation caused by the facility of Cu nanoparticles to agglomerate during the reaction process, which motivates the exploration for other kinds of catalysts.

Bimetallic- and intermetallic-based catalysts have proven to be efficient in terms of stability and hydrogen productivity for methanol steam reforming due to the cooperative effect between both metals. For instance, bimetallic PtNi catalysts are reported to achieve a good level of methanol conversion and high stability [15]. Several authors have demonstrated also that PdZn catalysts became highly active and selective for the reforming of methanol and it were attributed to the formation of PdZn alloys [16,17]. In a previous work, we have reported a catalyst based on supported NiSn nanoparticles (NPs) that is highly stable and active for the methanol steam reforming reaction [18]. We stated that the formation of surface alloys and the number of active sites exposed seems to be determinant for the optimal performance of this bimetallic systems.

In the last three decades, the most recent advances in catalysis science were essentially addressed in the study of nanocatalysis and *in situ-operando* characterization to precisely correlate the structure–reactivity relationship at the atomic or nanometer level [19]. Metal nanoparticles are defined by a highly reduced size in the range of 1–10 nm, and their geometric and electronic properties are intimately related to their catalytic performance. Inspired by our previous work [18], we have prepared a catalyst based on NiSn nanoparticles dispersed on alumina as support and we have compared the catalytic performance in methanol steam reforming with other analogous catalysts prepared by co-impregnation. In order to identify the active species and relate their formation with the surface catalyst properties as well as to obtain useful information for the preparation of more effective catalysts, we have performed an in situ diffuse reflectance infrared Fourier transform spectroscopy coupled to mass spectrometry (DRIFTS-MS) study of methanol adsorption under reaction conditions. Previous studies have demonstrated that DRIFTS is a powerful tool for monitoring catalytic surface chemistry before, during, and after adsorbate introduction [20]. This work provides information insights into the interaction of methanol with the surface of the catalyst and allows the elucidation of the reaction mechanism following the involved species during the reaction.

## 2. Materials and Methods

### 2.1. Chemical Reagents

Nickel (II) nitrate hexahydrate (Ni(NO_3_)_2_·6H_2_O, Panreac 99%), nickel acetate tetrahydrate (Ni(CH_3_COO)_2_·4H_2_O, Fluka 99%), anhydrous tin chloride (SnCl_2_, Fluka 97%), ethylenglycol (Panreac 99%), polyvinylpyrrolidone with an average molecular weight of 10,000 (PVP, Sigma-Aldrich 100%), sodium borohydride (Sigma-Aldrich 99%), acetone (Sigma-Aldrich 99%) and ethanol (Panreac 99%) were used as raw materials. Alumina powder (γ-Al_2_O_3_) used as support was obtained by the crushing of alumina spheres (Sasol, 1.78 mm diameter) in an agate mortar. 

### 2.2. Catalysts Synthesis

NiSn NPs were prepared by the polyol method using polyvinylpyrrolidone (PVP) as a protector agent as described elsewhere [21]. In a typical synthesis, the appropriate amounts of Ni(CH_3_COO)_2_·4H_2_O and SnCl_2_ precursor salts to achieve an atomic ratio Ni/Sn = 3 were dissolved in 70 mL of ethylenglycol (EG). Then, 0.7 g of PVP was added to the solution and heated to 50 °C while stirring until the formation of a completely homogenous solution. Afterwards, 0.34 g of NaBH_4_ was rapidly introduced for the reduction of the metallic salts. The obtained particles were refluxed at 200 °C for 3 h in order to assure the formation of Ni–Sn intermetallic compound. The black colloidal solution formed was then cooled down to room temperature and mixed with the appropriate amount of alumina in order to obtain 10 wt.% of NiSn nanoparticles supported over alumina. A small portion of the colloidal suspension was separated for HRTEM analysis. The solid was collected by centrifugation, washed several times with acetone and ethanol, and dried at 100 °C overnight. Finally, the sample was calcined at mild temperature of 350 °C for 1 h in order to remove the rest of PVP surrounding the nanoparticles. The obtained catalyst was labelled as NiSn-NPs.

Analogously, a similar catalyst was prepared by impregnation using Ni(NO_3_)_2_ and SnCl_2_ as precursor salts in the adequate amount to achieve an atomic ratio Ni/Sn of 3 and total loadings of 10 wt.%. In this case, the sample was calcined at 750 °C for 1 h and the catalyst obtained was designated as NiSn-IMP. The different calcination temperature selected in both catalysts is based on the TPR results as will be discussed below.

### 2.3. Characterization Methods

High-resolution transmission electron (HRTEM) micrographs were recorded on a 200 kV JEOL JEM-2010F instrument with a structural resolution of 0.19 nm at Scherzer defocus conditions. Scanning transmission electron microscopy (STEM) images were collected in the same instrument using a high-angle annular dark-field (HAADF) detector and an electron-beam probe of 0.5 nm to determine the particle size distribution.

In order to determine the crystalline structure of the prepared samples before and after reduction, X-ray diffraction (XRD) analysis was performed in a Siemens D500 diffractometer. Diffraction patterns were recorded using Cu *K**_α_* radiation (40 mA, 40 kV) and a position-sensitive detector using a step size of 0.05° and a step time of 1 s. 

Thermogravimetric analysis (TGA) was carried out on SDTQ60 thermobalance from 30 to 1200 °C with a heating rate of 2 °C min^−1^ by passing constant air flow of 50 mL min^−1^.

Temperature-programmed reduction (TPR) measurements were performed to study the reducibility of the prepared catalysts. The analyses were performed on 50 mg of fresh catalyst under 50 mL min^−1^ of a calibrated mixture of 3.85% H_2_ in Ar. The temperature was increased from room temperature up to 1000 °C at 15 °C min^−1^, and the H_2_ consumption was monitored by the TCD detector.

### 2.4. Catalytic Activity Tests

The methanol steam reforming catalytic tests were performed in a quartz fixed-bed reactor at atmospheric pressure. Prior to the reaction, the catalyst prepared by impregnation was reduced in situ at 800 °C for 1 h in 3 NmL min^−1^ of reductant flow (5% *v/v* H_2_:Ar). Then, the hydrogen flow was substituted by an 4Ar:1N_2_ mixture with a total flow of 17 NmL min^−1^ for purging hydrogen from reactant system and the temperature was decreased to the reaction one (350 °C). The pretreatment procedure for NiSn-NPs catalyst is identical but in this case the reduction temperature was 350 °C. After reduction pretreatment, the methanol–water mixture (1/2 molar ratio and 0.7 NL h^−1^ of mixture in gas phase) was added to the 4Ar:1N_2_ mixture entering the reactor. In all the experiments, the gas hourly space velocity (GHSV, STP conditions) was 26,000 h^−1^. The effluent stream was analyzed online using a gas micro-chromatograph equipped with two columns (Poraplot Q and molecular sieve 5 Å) and TCD detectors. The N_2_ in the stream feed inlet was used as internal standard. It should be mentioned that empty reactor and loaded with pure alumina showed no activity under these reaction conditions.

### 2.5. In Situ Drifts Measurements

In situ methanol adsorption studies followed by IR spectroscopy was performed using a high-temperature controlled DRIFT chamber from Spectra-Tech 101 with ZnSe windows coupled to a Thermo Nicolet Nexus infrared spectrometer using KBr optics and a MCT/B detector working at liquid nitrogen temperature. The sample was placed inside the chamber without packing or dilution, and the IR spectra were collected by accumulating 64 scans at 4 cm^−1^ resolution. Prior to methanol adsorption, the catalysts were pretreated at 300 °C for 1 h in a flow of 30 NmL min^−1^ of 5% H_2_ in helium. Methanol adsorption was performed at 100 °C for 15 min using a stream of He saturated with methanol vapor at room temperature. Then, pure He was passed through the sample and DRIFT spectra were recorded each 50 °C by increasing the temperature from 100 to 300 °C different temperatures. All the spectra were subtracted with a background spectrum recorded with an aluminum mirror. The effluent gases were analyzed on-line by mass spectrometry using a Balzers Omnistar system.

## 3. Results

### 3.1. General Characterization for Both Catalysts

Figure 1A,B show the TEM and HRTEM micrographs of the NiSn nanoparticles synthesized, respectively. As can be noticed from Figure 1A, it is evidenced the presence of well-dispersed nanoparticles together with bigger particles of undefined shape constituted by agglomeration of nanoparticles. Figure 1B shows that NiSn nanoparticles are formed by arranging small single crystals in different orientations presenting microdomains of sizes ranging from 2 to 4 nm. 

Figure 1C includes the HAADF image and the (energy dispersive X-ray) EDX analysis of a representative region of the NiSn nanoparticles having well dispersed as well as agglomerated particles. The chemical composition of the prepared NiSn nanoparticles shows that the material is quite heterogeneous, presenting particles with different compositions. The heterogeneity in the chemical composition could be related to the assembly of small particles with different compositions. It is noteworthy that the Sn concentration on the agglomerated particles is well above the nominal composition of the samples while the smaller nanoparticles present a lower tin concentration than nominal value. The EDX analysis points to the obtention of nanoparticles in which Ni_3_Sn, Ni_3_Sn_2_ and/or other Ni–Sn solid solution are coexisting. 

Figure 1D includes the histograms obtained for determining the average particle size of the prepared NiSn nanoparticles. The average size of the nanoparticles is found to be 2.3 ± 1.1 nm. In our previous work, we stated that the NiSn nanoparticle size is controlled by the amount of PVP added, where the coordination of PVP to the topmost surface atoms determines the growth rate of nanoparticles [21]. Based on this study, the amount of PVP used in the present work was adequate to obtain nanoparticles with an average size of 2 nm.

The prepared NiSn nanoparticles were dispersed onto alumina as detailed in the experimental procedure. In order to expose the nanoparticles completely, the capping agent (PVP) must be fully removed at the appropriated temperature to eliminate all the organic rest but also avoiding the metal nanoparticles sintering. This optimal temperature was determined by thermogravimetric analysis. Figure 2 shows the TG-DTA curve obtained for the as prepared NiSn-Nps catalyst. Firstly, a weak endothermic peak accompanied by a small weight loss are observed above 100 °C due to the desorption of physisorbed water. Around 200 °C, an exothermal peak emerges and simultaneously occurs a fast weight loss revealing that organic matter starts to be burned. The combustion process is completed at around 400 °C. Note that the other two small exothermic peaks accompanied by the slight increase of mass were observed between 400 and 600 °C, which indicates that metal nanoparticles become oxidized in air at this temperature. Based on these results, the NiSn-NPs catalyst was calcined in air at 350 °C.

Figure 3 displays the TPR profiles obtained for both calcined catalysts. As can be observed, the catalyst prepared by impregnation exhibits three main reduction peaks at around 437 °C, 570 °C and 798 °C, which are related to the reduction of nickel phases with a different morphology and/or degree of interaction with the support. According to the literature, the reduction peak at around 437 °C corresponds to the reduction of large crystallites of nickel oxide weakly interacting with the support while the peak at 570 °C may be ascribed to the reduction of smaller crystallites of nickel oxide [22]. Meanwhile, the reduction peak at higher temperature (798 °C) is mostly related to the reduction of nickel incorporated into the NiAl_2_O_4_ structure, which presents a strong interaction with the support [23]. Concerning the NiSn-Nps catalyst, it can be noted that there only appears one reduction peak at 374 °C. The presence of this low-temperature peak suggests that a thin passive layer of NiO was formed when the supported NiSn nanoparticles were exposed to atmospheric air in agreement with our previous work [18]. 

The X-ray diffraction patterns of both catalysts calcined and after TPR reduction tests are shown in Figure 4. As can be seen, both fresh catalysts show the diffraction lines at 66.32, 45.47 and 37.36° associated to the alumina bare support. Note that the NiSn-Nps fresh catalyst shows several reflections related to nanosized NiSn alloys unrevealing that nanoparticles are dispersed onto alumina support. The presence of NiSn intermetallic compounds and the low-temperature calcination (350 °C) inhibits the formation of other phases such as NiAl_2_O_4_. By contrast, diffraction lines corresponding to NiO species (43.30 and 62.90°) were hardly observed in NiSn-IMP fresh catalyst suggesting that surface nickel aluminate is formed as clearly evidenced the slight shift towards smaller 2θ angles in diffraction lines of alumina. Diffraction lines associated to Sn and/or NiSn phases either were detected in the catalyst prepared by impregnation likely due to the high dispersion or amorphous character of tin in this sample. These observations are in concordance with our previous results already published [18,24].

In order to determine the structural changes produced during the reduction process, X-ray diffraction analyses after TPR tests were also performed. As shown in Figure 4, it is remarkable that the XRD pattern of reduced NiSn-IMP catalyst shows the appearance of three intense diffraction peaks at 44.5, 51.9 and 76.4° which are ascribed to metallic nickel (Ni^0^) [25]. In addition, it can be observed that there is the presence of low intense reflections that may be related to NiSn intermetallic compounds [18,26]. Furthermore, it is noteworthy that diffraction lines of alumina (66.32, 45.47 and 37.36°) were shifted to higher 2θ angles after TPR evidencing that Ni^2+^ ions were rejected from NiAl_2_O_4_ spinel being reduced into metallic nickel and extracting Al_2_O_3_. Therefore, large amounts of metallic nickel and the rest of the NiSn alloys are coexisting in the reduced NiSn-IMP catalyst after reduction at temperatures above 750–800 °C. Concerning the NiSn-Nps reduced catalyst, the appearance of well-defined and intense diffraction peaks after the reducibility test clearly evidences that NiSn nanoparticles were agglomerated into large crystals of NiSn alloys compounds with increasing the temperature above 600 °C during TPR run test. Note that for methanol reforming tests this catalyst was activated at 350 °C and at this reduction temperature the nanoparticles remain highly dispersed onto alumina support.

### 3.2. Catalytic Performance Testing on Methanol Steam Reforming

Figure 5 shows the methanol conversion and the H_2_/CO molar ratio as a function of the reaction temperature obtained for both of the two catalysts tested in methanol steam reforming. It can be observed in Figure 5A that methanol conversion increases with the temperature for both catalysts and the catalyst prepared by impregnation (NiSn-IMP) is more active than the catalyst based on supported nanoparticles (NiSn-Nps). These differences in activity can be explained from the structure of the active phase discussed above. The catalyst prepared by impregnation was activated at 800 °C and the NiAl_2_O_4_ spinel phase was mainly reduced to metallic Ni forming only a small amount of NiSn alloy. Meanwhile, the catalyst based on NiSn nanoparticles was reduced at 350 °C and the active phase is essentially constituted of nanosized NiSn alloy. As Ni metallic is much more active for cleavage C–H bonds than NiSn alloy, it can be deduced that higher methanol conversion will be achieved in NiSn-IMP catalyst. According to the results of our previous studies [18], NiSn nanoparticles based catalysts are less active for methanol steam reforming but notably increase the catalytic stability and inhibit the accumulation of carbonaceous deposits. 

On the other hand, Figure 5B shows that H_2_/CO molar ratio decreases with increasing the reaction temperature. It is expected that since the reverse water gas-shift reaction will become more favorable than the water gas-shift reaction (Equation (3)) with increasing the reaction temperature. Nevertheless, it is noteworthy that the NiSn-Nps catalyst outperforms NiSn-IMP in terms of H_2_/CO molar ratio. This demonstrates that the presence of NiSn nanoparticles was effective in suppressing CO formation. Hydrogen-rich fuel stream are particularly important for hydrogen supply to PEMFC’s fuel cells, which require CO concentrations lower than 100 ppm levels [27]. Typically, Ni-based catalysts are associated to higher CO production due to its inherent selectivity towards methanol decomposition (Equation (2)) [28]. However, our results demonstrate that NiSn nanoparticles changes the reaction mechanism. This minor CO production observed for NiSn-Nps catalyst could be attributed to the particular electronic properties of the NiSn nanoparticles as active centers for methanol steam reforming. Therefore, this catalytic behavior clearly evidences that small nanosized NiSn particles modifies the catalytic properties of the solid in comparison to the analogous catalytic system prepared by impregnation and demonstrate good potential use for methanol steam reforming reaction.

### 3.3. In Situ DRIFTS-MS Methanol Adsorption Studies

To further evaluate the differences observed in catalytic activity, we performed an in situ DRIFT spectroscopic study of methanol adsorption to elucidate the intermediates involved in each catalyst during the methanol reforming reaction. 

Figure 6 includes the IR spectrum collected after the saturation of methanol at 100 °C onto the NiSn-IMP catalyst. The emerging bands correspond to the surface species that are formed during methanol adsorption while the vanishing features are related to the hydroxyl surface interacting with methanol during the adsorption process. In fact, the bands at 3736 and 3690 cm^−1^ can be ascribed to terminal and bridge hydroxyl groups, respectively, according to the model of Knözinger [29]. According to this model, five types different of OH groups can be distinguished in alumina, and every one of them presents a ν(OH) vibrational stretching active mode. In our case, methanol adsorption only affects two types of OH: (i) terminal hydroxyl, in which oxygen is bonded to one metallic atom of the surface, and (ii) bridge hydroxyl with an oxygen bridged to two metallic atoms. Moreover, it can be noticed in Figure 6 the appearance of a broad band centred at around 3300 cm^−1^, which is assigned to hydroxyl groups interacting by hydrogen bonds with physisorbed methanol [30].

Regarding the adsorbed species formed after methanol adsorption, it is clearly evident that there is the appearance of two intense bands at 1059 and 1000 cm^−1^ ascribed to the ν(CO) stretching modes of methoxide groups [31]. The appearance of more than one band in this region is related to the presence of methoxide species with different coordination. According to Lamotte et al. [32], it can be stated that the band at 1059 cm^−1^ corresponds to the ν(CO) vibrational mode of monodentate methoxides while the band at 1000 cm^−1^ can be ascribed methoxide with coordination bridge-type. This attribution is based on the type of OH groups affected by the adsorption of the alcohol (monodentates and bridge). On the other side, the positive band of very low intensity observed at 3658 cm^−1^ corresponds to adsorbed methanol species without dissociating and this band is associated to another broad band which emerged at 1440 cm^−1^. Furthermore, in the 3100–2700 cm^−1^ region appears a complex set of overlapped bands ascribed to the ν(C-H) vibration modes of methyl groups coming from dissociated and no-dissociated methanol adsorbed species [30]. Finally, the band that appears at 2070 cm^−1^ can be assigned to the presence of CO linearly adsorbed on Ni sites [33]. Although this band is low intense, it reveals that this catalyst is able to produce CO at 100 °C, which is a by-product of steam reforming of methanol. The simultaneous production of CO and H_2_ resulting of methanol decomposition during the adsorption step is already proposed by some authors [31].

Figure 7 shows the DRIFT spectrum recorded after the saturation of methanol at 100 °C onto NiSn-Nps catalyst. In this case, the observed bands identical than that of NiSn-IMP sample, and thus all of them can be attributed to the same species, but an important difference was observed. In contrast to NiSn-IMP, the intensity of the band related to monodentate methoxide species (1057 cm^−1^) is more major than that of one associated to bridge methoxide species (1009 cm^−1^). This indicates that the preparation method based on deposition of NiSn nanoparticles leads to minor concentration of surface bridge hydroxyls over the surface and consequently it does affect directly to the methoxide intermediate species formed during methanol adsorption. 

After saturation of catalyst surface with methanol at 100 °C, the sample was purged with helium flow, maintaining the temperature during 5 min, and subsequently the temperature was progressively increased up to 300 °C. In order to clearly appreciate the thermal evolution of the surface species, the spectrum recorded at each temperature was subtracted from a reference spectrum collected after methanol saturation. Thus, the negative bands correspond to species that are disappearing during the thermal treatment whereas the positive ones represent the formed species at each temperature. Moreover, the effluent stream was monitored by MS to correlate the adsorbed surface species with the desorbed products. It must be pointed out that the measured MS signals only give a qualitative idea of the evolution of different products.

Figure 8 illustrates the thermal evolution of the adsorbed species on NiSn-IMP catalyst. As can be observed, at 200 °C there takes place the disappearance of both bands at 1059 and 1000 cm^−1^ related to the ν(CO) vibration modes of the methoxide species adsorbed with different coordination geometry. Additionally, it is also noticed that features at 2800–3000 cm^−1^ related to ν(CH) vibration modes and the band at 2070 cm^−1^ typical of CO-linearly adsorbed on nickel also disappeared at 200 °C. Moreover, it is noteworthy that bands attributed to species linked by hydrogen bonds (3300 and 1450 cm^−1^) vanished and the features of surface hydroxyl groups emerged simultaneously. On the other hand, it can be noticed that as temperature increased a new band appeared at 1590 cm^−1^. This band is related to formate intermediates, and their formation indicates that methoxide species are transformed into formates. It is worth mentioning that formate species are characterized by three bands at 1590, 1390 and 1370 cm^−1^, which correspond to the ν_as_(COO) asymmetric COO stretching mode, the δ(CH) out-of-plane bending CH mode and the ν_s_(COO) symmetric COO stretching mode, respectively [34].The bands at 1390 and 1370 cm^−1^ are only detected at 300 °C because they are masked by the broad negative band around 1440 cm^−1^ at lower temperatures.

Figure 9 shows the evolution of the effluent stream followed by MS during the thermal treatment. At 100 °C, it can be observed the release of water, CO, methanol and hydrogen, which is in concordance with the adsorbed species detected in the DRIFT spectrum (Figure 7). This reveals that this catalyst can produce H_2_ and CO from methanol at 100 °C. On the other side, it is worth mentioning that the transformation of methoxide species in formates can produce hydrogen with the simultaneous surface reduction of the support since at 200 °C no modification was observed in the intensity of the formates DRIFT bands (Figure 7) and the *m/z* signal of hydrogen falls to zero in the MS (Figure 8). In a first approach, we can suggest that part of the produced hydrogen can be due to the transformation of methoxides into formates via reduction of support, although the confirmation of this fact requires further studies. As temperature was increased at 200 °C, dimethyl ether (DME) was detected in the stream effluent. Carrizosa et al. [35] proposed a mechanism for explaining the formation of dimethyl ether by a reaction between adsorbed methoxy species during methanol adsorption on anatasa. Accordingly, the surface reaction occurs when a certain coverage of the so-called labile alcoholates is achieved and it requires the presence of acid sites. We believe that these conditions are satisfied in the NiSn-IMP catalyst when desorption temperature was increased.

Figure 10 displays the thermal evolution of the adsorbed species on the NiSn-Nps catalyst. It must be emphasized that notable differences are observed in comparison with the impregnated catalyst. Indeed, as the temperature is increased, the diminution of the band at 1057 cm^−1^ related to monodentate methoxides is much faster than that of band at 1009 cm^−1^ ascribed to bridge-coordinated methoxides. This observation is in concordance with the lower thermal stability of monodentate methoxides. Further, it should be noted that a slight increase of the bands at 1030 and 1009 cm^−1^ occurs under helium flow at 100–200 °C, which can be attributed to the presence of non-dissociated methanol adsorbed (1030 cm^−1^) and bridge-coordinated methoxides (1009 cm^−1^). Remarkably, the appearance of these two bands is accompanied by a slight decrease of the band around 3700 cm^−1^. The increase of the band at 1009 cm^−1^ and the simultaneous diminution of the band at 3700 cm^−1^ reveal the presence of bridge-coordinated methoxides formed from the adsorbed methanol. As mentioned above, bridge methoxides are thermally more stables than monodentate ones. As the temperature was increased above 200 °C, the most stables bridge methoxides disappeared and three new bands at 1596, 1390 and 1370 cm^−1^ ascribed to formate species emerged simultaneously [34]. In comparison with NiSn-IMP catalysts, the formation of formate species occurs more slowly in this case and besides the intensity of the bands increased with the temperature.

Figure 11 shows the evolution of mass spectrometer signal intensities recorded as a function of time and temperature during the desorption of adsorbed methanol on NiSn-Nps catalyst. It should be noted that methanol was desorbed even at temperatures above 200 °C. In concordance with the DRIFT spectra shown in Figure 10, it can be assumed that methoxide species weakly adsorbed (monodentate geometry) are transformed into methanol at 150–200 °C. This suggestion is in agreement with the data reported by Lamotte et al. [32]. Moreover, it explains the presence of non-dissociated methanol still adsorbed at 200 °C. Meanwhile, the desorbed methanol detected in NiSn-IMP sample at 100–150 °C is essentially attributed to rests of physisorbed methanol (see Figure 9). 

On the other side, it can be observed that NiSn-Nps catalyst also produces CO and H_2_ by methanol decomposition between 100 and 200 °C. This indicates that the NiSn-Nps catalyst also presents an optimal activity for steam reforming of methanol. However, differently to NiSn-IMP, it must be remarked that hydrogen was also produced at temperatures above 300 °C. This suggests that NiSn nanoparticles facilitates the decomposition of formates into hydrogen and CO_2_ and explain the higher hydrogen production observed in NiSn-Nps catalyst in catalytic tests of methanol steam reforming. Finally, Figure 10 indicates that DME was also produced in NiSn-Nps catalyst above 200 °C but their formation was more prolongated with time-on-stream. This fact suggests that NiSn-Nps catalyst contains superficial species, likely a higher concentration of strong acid sites, that leads to the formation of DME methanol dehydration. Notwithstanding, this assertion requires further studies of acid-base properties and herein it is only indirectly assumed. 

### 3.4. Mechanistic Insights

From a mechanistic point of view, the experimental results of this work show that the production of hydrogen by methanol steam reforming is sensitively affected by size of active metal. From an in situ DRIFTS-MS study, it can de deduced that the adsorption of methanol on the catalyst is mainly dissociative and methoxy intermediates with different geometry are formed on the active sites and over inert regions of the catalyst surface mainly associated to alumina. Our results suggest that part of these methoxy groups, likely those located at the metal–support interface or on metal sites, transform easily to formate species, which subsequently are decomposed into hydrogen and CO_2_. Some authors have proposed that the transformation of methoxy groups into formate species occurs via intermediacy of dioxymethylene species [36]. However, these intermediates are highly unstable and cannot be detected by DRIFTS during the reaction process. Figure 12 illustrates a tentative reaction pathway for the methanol steam reforming process. As mentioned in the introduction section, the steam reforming reaction is formally the sum of methanol decomposition and water gas shift reaction. Accordingly, CO produced by methanol decomposition can be transformed into CO_2_ and H_2_ via WGS reaction likely also through the formation of formate intermediates at the interface sites. The presence of steam during methanol decomposition displaces the production of CO towards hydrogen and CO_2_. The reaction scheme also includes methoxy condensation to DME that probably involves the acid sites of alumina, being a well-known acid-catalyzed reaction [36,37]. On the basis of the obtained results, it seems reasonable to assume that the reaction mechanisms for methanol steam reforming in both catalysts are closely related, and that the surface phenomena occurring are very similar. However, NiSn nanoparticles dispersed onto alumina increases the number of metal–support interface sites exposed and favor the formate decomposition increasing the hydrogen yield.

## 4. Conclusions

This study provides new insights into the reaction mechanism of methanol steam reforming for hydrogen production both supported NiSn nanoparticles and analogous NiSn catalyst prepared by impregnation. In situ DRIFTS-MS measurements revealed that different intermediate species such as methoxides with different geometry (bridge and monodentate) and formate species are the most involved in the methanol steam reforming reaction in both catalysts. Our results point out that methoxy groups mainly located at the metal–support interface are more easily transformed into formate species, which subsequently are decomposed into hydrogen and CO_2_. Consequently, the major presence of metal–support interface sites in supported NiSn Nps leads to a higher hydrogen yield. 

## Figures and Tables

**Figure 1 nanomaterials-11-03234-f001:**
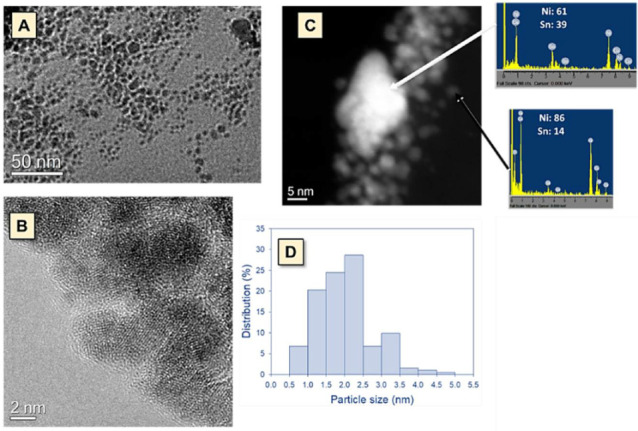
TEM (**A**) and HRTEM (**B**) micrographs of synthesized NiSn nanoparticles, EDX analysis of a representative region of the NiSn nanoparticles having well dispersed as well as agglomerated particles (**C**), and distribution of NiSn particle sizes derived by TEM images (**D**).

**Figure 2 nanomaterials-11-03234-f002:**
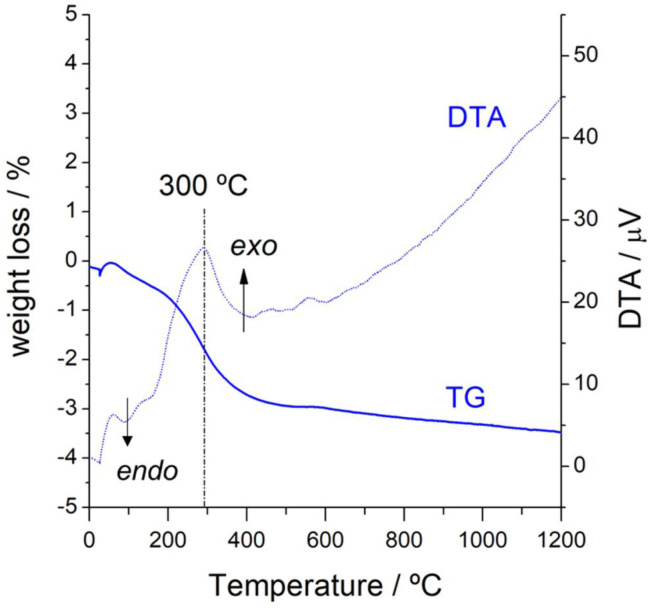
TG-DTA curves of the as prepared NPs-based catalyst (NiSn−NPs).

**Figure 3 nanomaterials-11-03234-f003:**
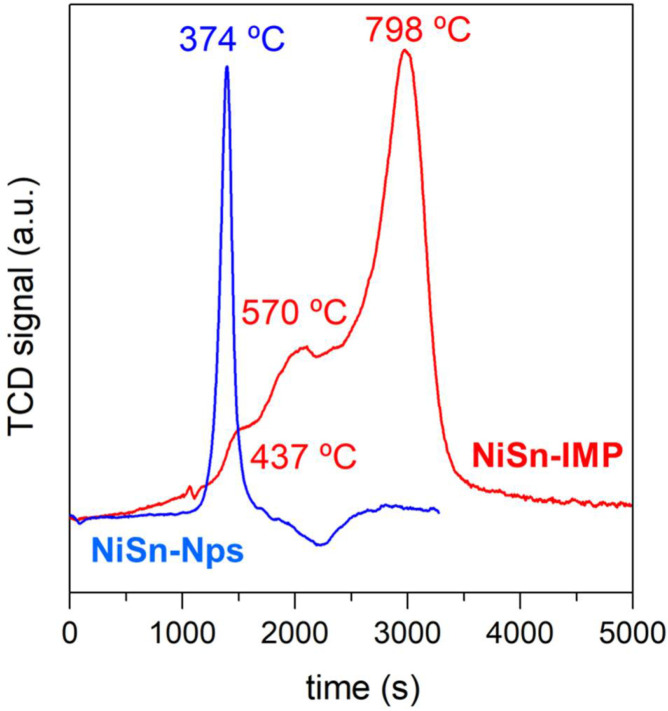
H_2_-TPR profiles of both calcined catalysts.

**Figure 4 nanomaterials-11-03234-f004:**
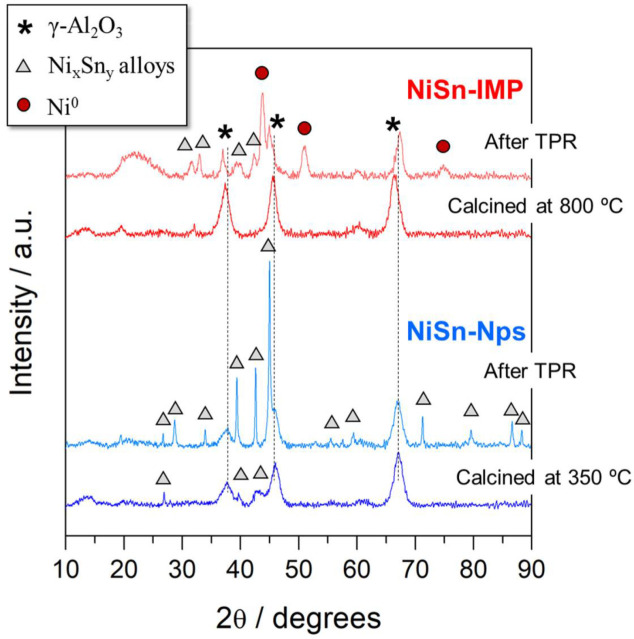
XRD patterns of both NiSn−IMP and NiSn−Nps catalysts calcined and after TPR measurements.

**Figure 5 nanomaterials-11-03234-f005:**
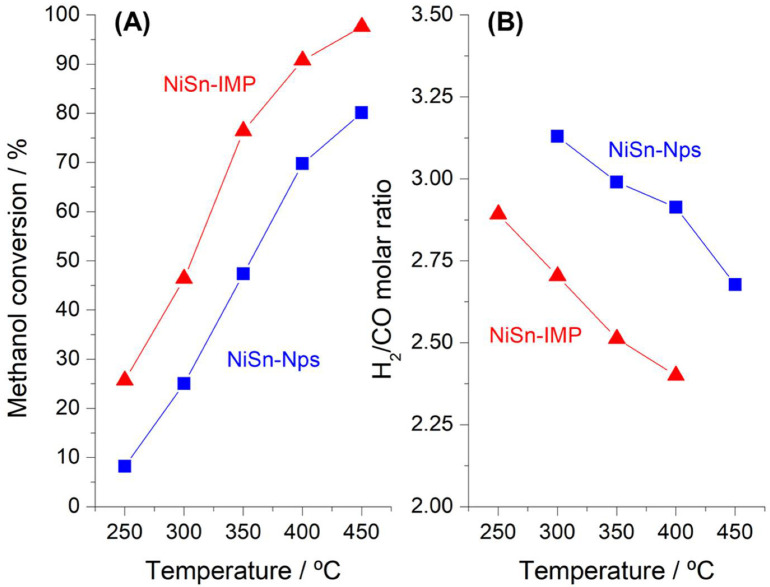
Methanol conversion (**A**) and H_2_/CO molar ratio (**B**) as a function of the temperature for the NiSn−IMP and NiSn−Nps catalysts.

**Figure 6 nanomaterials-11-03234-f006:**
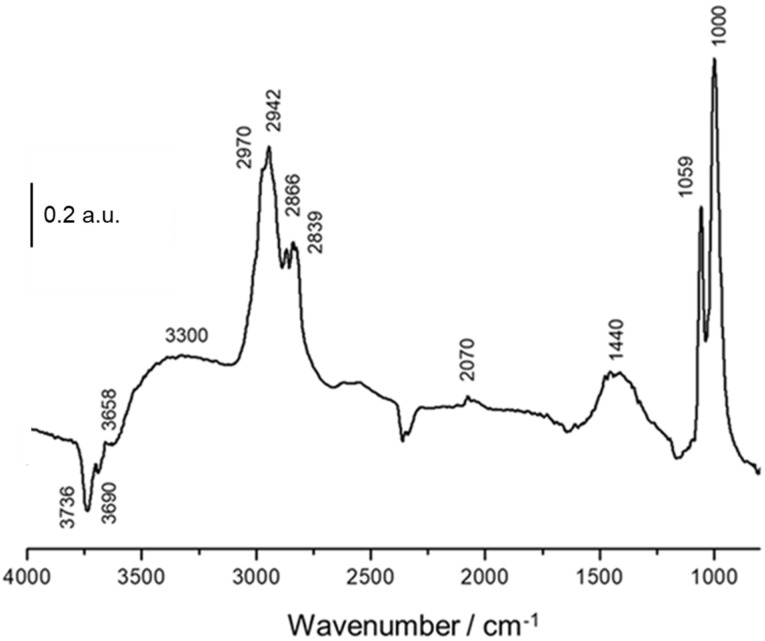
DRIFT spectra of NiSn−IMP catalyst after adsorption of methanol at 100 °C.

**Figure 7 nanomaterials-11-03234-f007:**
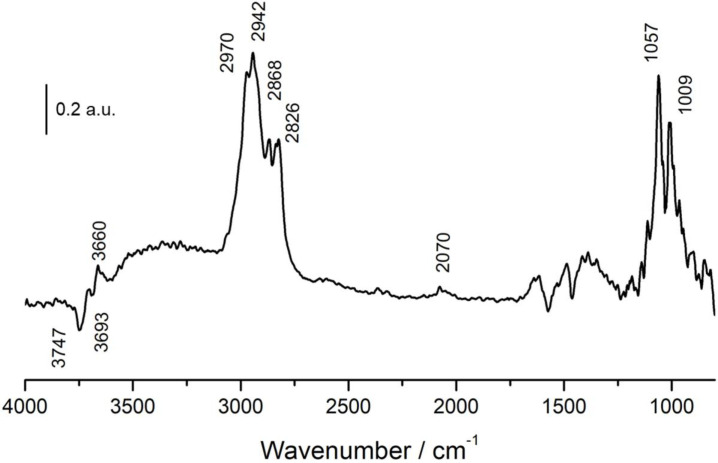
DRIFT spectra of NiSn−Nps catalyst after adsorption of methanol at 100 °C.

**Figure 8 nanomaterials-11-03234-f008:**
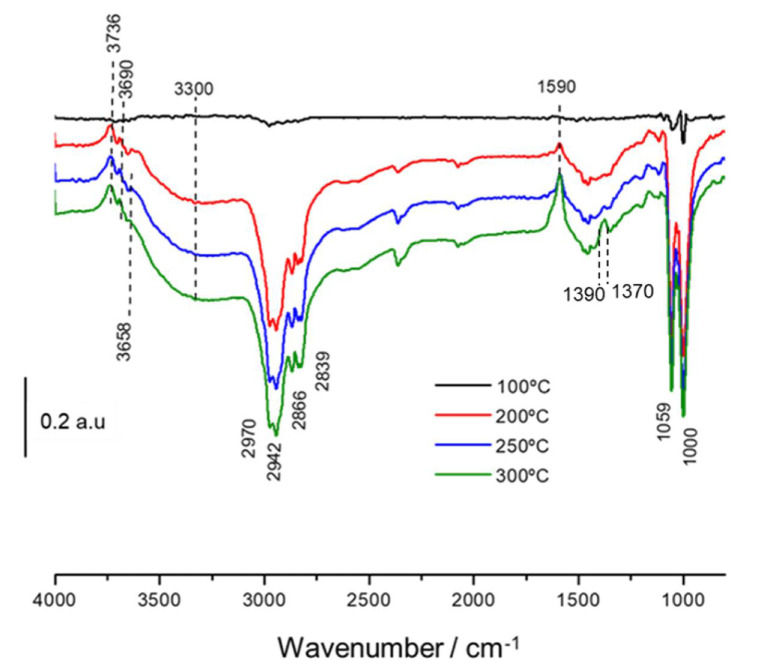
Thermal evolution of difference spectra after methanol adsorption at 100 °C under flow of helium at 100 °C, 200 °C, 250 °C and 300 °C on NiSn−IMP catalyst.

**Figure 9 nanomaterials-11-03234-f009:**
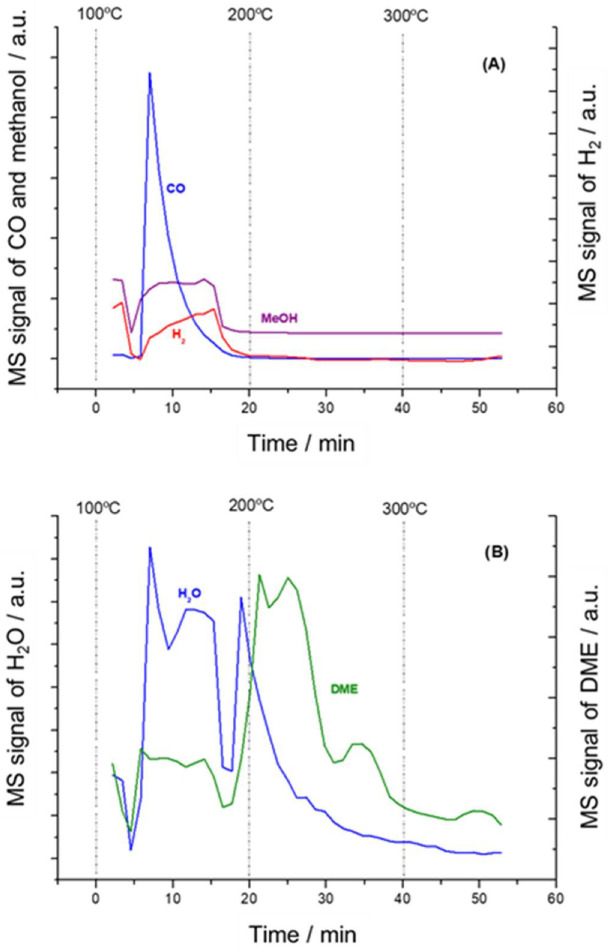
Evolution of MS signal intensities recorded as a function of time and temperature during the desorption of adsorbed methanol over NiSn-IMP sample: (**A**) CO, H_2_ and methanol; (**B**) H_2_O and DME.

**Figure 10 nanomaterials-11-03234-f010:**
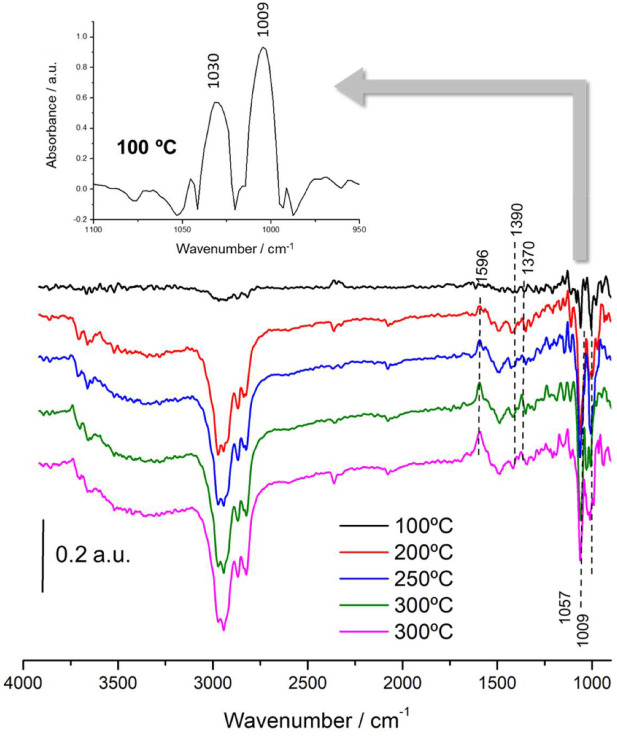
Thermal evolution of difference spectra after methanol adsorption at 100 °C under flow of helium at 100 °C, 200 °C, 250 °C and 300 °C on NiSn−Nps catalyst.

**Figure 11 nanomaterials-11-03234-f011:**
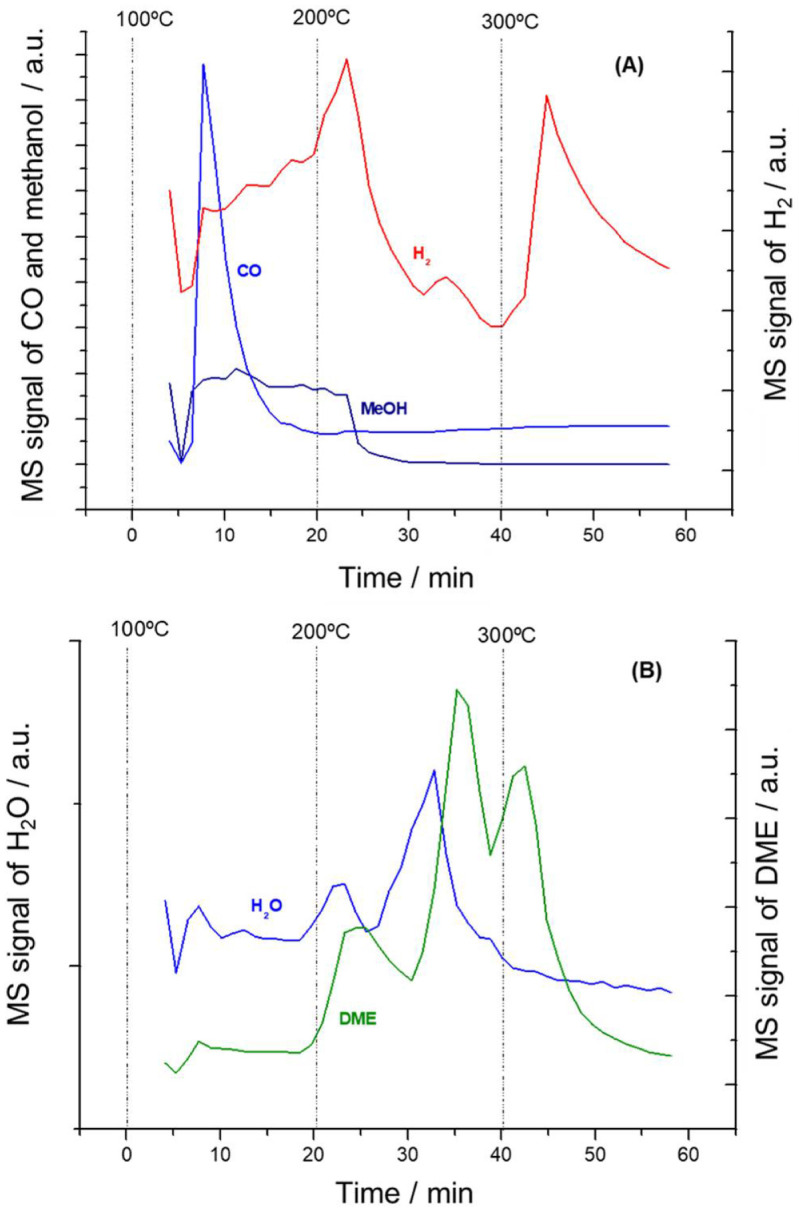
Evolution of MS signal intensities recorded as a function of time and temperature during the desorption of adsorbed methanol over NiSn−Nps sample. (**A**) CO, H_2_ and methanol; (**B**) H_2_O and DME.

**Figure 12 nanomaterials-11-03234-f012:**
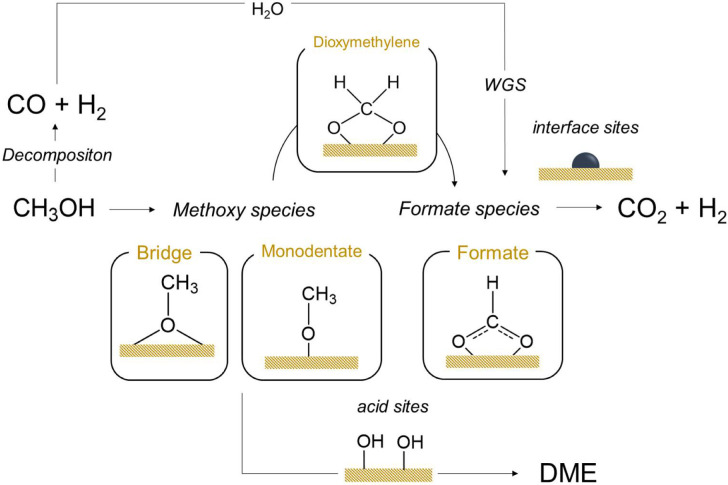
Tentative reaction mechanism proposed for methanol steam reforming reaction.

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
