# Peer review of "In Situ DRIFTS-MS Methanol Adsorption Study onto Supported NiSn Nanoparticles: Mechanistic Implications in Methanol Steam Reforming"

_nanomaterials, 2021, doi:10.3390/nano11123234_

Round 1

Reviewer 1 Report

In this work, authors reported some significant results on studying the reaction mechanism of methanol steam reforming for hydrogen production on supported NiSn nanoparticles catalysts by combining in situ DRIFTS-MS measurements. The whole manuscript is well organized, and might be accepted for publication after the following questions are addressed.

  • In Figure 8, an obvious FT-IR peak appears at around 1330 cm-1, according to some related literatures (see below), this signal together with the signal at around 1600 cm-1 might be assigned to the adsorbed bidentate formate. Please pay attention to the explanation on the reason why the two peaks at 1370 and 1390 cm-1 are absent, since it was proposed that the peak appeared at around this position is related to the adsorbed unidentate formate.
  • For comparison, Figure 10 should be present in the same style as Figure 8, better including the whole wavenumber range. Besides, there is a weak signal at around 1650 cm-1, please check if its intensity has a relationship with that of 1370 cm-1, since these two signals have been assigned to the stretching vibrations of C=O and C-O in unidentate formate.   
  • Line 381, formats should be formates.

References:

  1. Unland, M.L. Infrared study of methanol decomposition on alkali metal X-type zeolites. J. Phys. Chem. 1978, 82, 580–583.
  2. King, S.T.; Garces, J.M. In situ infrared study of alkylation of toluene with methanol on alkali cation-exchanged zeolites. J. Catal. 1987, 104, 59–70.
  3. Zhang, Z.H.; Yuan X.L.; Miao S.S.; Li H.; Shan W.L.; Jia M.J.; Zhang C.L. Effect of Fe Additives on the Catalytic Performance of Ion-Exchanged CsX Zeolites for Side-Chain Alkylation of Toluene with Methanol, Catalysts, 2019, 9, 829.

Author Response

We appreciate the reviewers’ constructive comments. These proved to be very helpful in improving the quality of our manuscript, and we have earnestly addressed all of them in the revised submission.

Reviewer 1

In this work, authors reported some significant results on studying the reaction mechanism of methanol steam reforming for hydrogen production on supported NiSn nanoparticles catalysts by combining in situ DRIFTS-MS measurements. The whole manuscript is well organized, and might be accepted for publication after the following questions are addressed.

  • In Figure 8, an obvious FT-IR peak appears at around 1330 cm-1, according to some related literatures (see below), this signal together with the signal at around 1600 cm-1might be assigned to the adsorbed bidentate formate. Please pay attention to the explanation on the reason why the two peaks at 1370 and 1390 cm-1 are absent, since it was proposed that the peak appeared at around this position is related to the adsorbed unidentate formate.

The bands at 1370 and 1390 cm-1 are present and they are remarkably notable at 300 ºC. We have marked these bands in Fig.8 and we can assign to formate species. However, these bands are masked at lower temperatures by a broad band centered at 1440 cm-1 related to adsorbed methanol. Accordingly, we have included these comments in the text.

  • For comparison, Figure 10 should be present in the same style as Figure 8, better including the whole wavenumber range. Besides, there is a weak signal at around 1650 cm-1, please check if its intensity has a relationship with that of 1370 cm-1, since these two signals have been assigned to the stretching vibrations of C=O and C-O in unidentate formate.   

We have modified the Figure 10 including the whole range for better comparison with Figure 8. The weak signal es related to the formation of water during the dehydration process of methanol while band at 1370 cm-1 is related to formates species.

  • Line 381, formats should be formates.

As referee indicates, this mistake was corriged

References:

  1. Unland, M.L. Infrared study of methanol decomposition on alkali metal X-type zeolites. J. Phys. Chem. 1978, 82, 580–583.
  2. King, S.T.; Garces, J.M. In situ infrared study of alkylation of toluene with methanol on alkali cation-exchanged zeolites. J. Catal. 1987, 104, 59–70.
  3. Zhang, Z.H.; Yuan X.L.; Miao S.S.; Li H.; Shan W.L.; Jia M.J.; Zhang C.L. Effect of Fe Additives on the Catalytic Performance of Ion-Exchanged CsX Zeolites for Side-Chain Alkylation of Toluene with Methanol, Catalysts2019, 9, 829.

Reviewer 2 Report

This manuscript provides information insights into the interaction of methanol with the surface of the catalyst and investigates the reaction mechanism. I found the manuscript clearly written and easily understandable. The manuscript contains original (unpublished before) and useful information which may be interesting to some of the Nanomaterials readers. Therefore, I recommend publication of the review after addressing the following comments:

The abstract needs improvement. Please add more information on research results. It should be informative and completely self-explanatory.

Abstract: It’s not common to start the abstract referring to the previous work. Please revise the sentence “On the basis of our previous investigations, we have demonstrated…”.

SEM images will help to gain further information on the morphology of the synthesized catalysts.

I would recommend improving the quality of DRIFT spectra (Figs. 6 and 7).

Have the authors investigated the stability of the catalyst in successive methanol steam reforming catalytic tests?

Some of the references (especially those in the introduction part) should be replaced with recent references such as:
ACS Applied Materials & Interfaces, 13 (2021) 13072-13086.
Applied Surface Science, 554 (2021) 149518.
Journal of Cleaner Production, 275 (2020) 124157.
Renewable Energy, 182 (2022) 713-724.

Author Response

Reviewer 2:

This manuscript provides information insights into the interaction of methanol with the surface of the catalyst and investigates the reaction mechanism. I found the manuscript clearly written and easily understandable. The manuscript contains original (unpublished before) and useful information which may be interesting to some of the Nanomaterials readers. Therefore, I recommend publication of the review after addressing the following comments:

  • The abstract needs improvement. Please add more information on research results. It should be informative and completely self-explanatory.
  • Abstract: It’s not common to start the abstract referring to the previous work. Please revise the sentence “On the basis of our previous investigations, we have demonstrated…”.

We have modified the abstract as reviewer proposes.

  • SEM images will help to gain further information on the morphology of the synthesized catalysts.

We agree with the referee. However, we don´t expect significant changes in the morphology of the catalyst and/or the alumina support. It is most relevant the anslysis of NiSn nanoparticles by HR-TEM as shown in this work.

  • I would recommend improving the quality of DRIFT spectra (Figs. 6 and 7).

As referee kindly suggests, we have improved the resolution of Figs. 6 and 7

  • Have the authors investigated the stability of the catalyst in successive methanol steam reforming catalytic tests?

The stability catalytic of these samples was reported in a previous work (Int. J. Hydrogen Energy 38, 2013, 6646—6656)

  • Some of the references (especially those in the introduction part) should be replaced with recent references such as:

ACS Applied Materials & Interfaces, 13 (2021) 13072-13086.
Applied Surface Science, 554 (2021) 149518.
Journal of Cleaner Production, 275 (2020) 124157.
Renewable Energy, 182 (2022) 713-724.

As referee proposes we have included more recent references in the introduction section and discussed about it.
